# Mutual Information-Guided Corruption for Improved Self-Supervised Representation Learning in Tabular Data

**Michael Lawson** [* 1]  **Emerald Sy** [* 1]  **Zhang Kehui** [* 1]  **Raymond H. F. Chan** [1 2]  **Kannie W. Y. Chan** [1 3]
**Rosa H. M. Chan** [1 3]

## Abstract

Self-supervised learning has revolutionised representation learning in computer vision and natural language processing, yet tabular data remains challenging due to heterogeneous feature distributions and complex inter-feature dependencies. Whilst recent methods use random feature corruption for pretraining, they typically ignore the statistical structure inherent in tabular datasets. We propose a self-supervised learning framework that leverages mutual information to automatically discover and exploit feature dependencies during pretraining. Our approach constructs feature groups based on statistical relationships and uses these to guide data augmentation, by incorporating conditional variational autoencoders for realistic sample generation. Experiments on plaque prediction from pretraining on UK Biobank data, 6 open-source medical datasets, and 69 OpenML-CC18 benchmark tasks demonstrate superior label efficiency, requiring fewer labeled examples to reach comparable performance.

## 1. Introduction

The success of deep learning has been underpinned by the availability of large-scale labelled datasets in domains such as computer vision and natural language processing. However, obtaining labelled data remains expensive and time-consuming across many application areas, particularly in specialised domains like healthcare, finance, and scientific research. Self-supervised learning has emerged as a compelling solution to this challenge by enabling models to learn meaningful representations from unlabelled data through carefully designed pretext tasks (Jing & Tian, 2020).

Contrastive learning has become one of the most successful self-supervised paradigms, demonstrating that models can learn powerful representations by maximising agreement between differently augmented views of the same data whilst minimising agreement between views of different data points. SimCLR (Chen et al., 2020) showed that simple data augmentations combined with a contrastive loss could produce representations that rival or exceed supervised pretraining for image classification. MoCo (He et al., 2020) introduced a momentum-based approach that maintains a queue of negative examples, enabling efficient contrastive learning at scale.

The extension of self-supervised learning to tabular data (Darabi et al., 2021; Zhang et al., 2023) presents a challenge, as tabular data possess a rudimentary structure that does not inherently carry additional meaning. Unlike images, where spatial augmentations like cropping and rotation preserve semantic meaning (Krizhevsky et al., 2012; He et al., 2016), or text, where masking and reordering maintain linguistic structure (Devlin et al., 2019; Raffel et al., 2020; Song et al., 2019), tabular data lacks such obvious invariances. Features in tabular datasets are often heterogeneous: spanning continuous measurements, categorical variables, and ordinal scales. Moreover, the relationships between features are typically dataset-specific and may not follow intuitive patterns. Recent work has begun to address these challenges through various approaches. VIME (Yoon et al., 2020) introduced a pretext task based on mask reconstruction combined with feature vector estimation. SubTab (Ucar et al., 2021) proposed generating subsets of features as different views for contrastive learning. TabPFN (Hollmann et al., 2022; 2025) took a different approach by training a transformer to perform approximate Bayesian inference over synthetic tabular datasets, enabling few-shot learning without task-specific fine-tuning.

Despite these advances, current state-of-the-art methods for tabular self-supervised learning largely treat features as independent entities or rely on manual specification of feature relationships. This approach overlooks a fundamental characteristic of many real-world tabular datasets: features often

---

[*]Equal contribution   [1]Hong Kong Centre for Cerebrocardiovascular Health Engineering, Hong Kong [2]Lingnan University, Hong Kong [3]City University of Hong Kong, Hong Kong. Correspondence to: Michael Lawson <mymclawson@hkcoche.org>.

*Proceedings of the Workshop on Foundation Models for Structured Data at the 43rd International Conference on Machine Learning*, Seoul, South Korea. Copyright 2026 by the author(s).

exhibit strong dependencies that encode important information. In healthcare data, for instance, related biomarkers tend to covary in ways that reflect underlying physiological processes. Ignoring or discarding these dependencies during pretraining may result in representations that fail to capture the full structure of the data distribution.

Our work is most similar to SCARF (Bahri et al., 2022), which generates views by randomising features independently, using the InfoNCE loss (Gutmann & Hyvärinen, 2010; Oord et al., 2018) for pretraining. Whilst SCARF represents an important advance in tabular self-supervised learning, it treats features as independent entities and does not explicitly model or explore the statistical dependencies between features. Mutual information (Shannon, 1948) has previously been applied to handle heterogeneous tabular data in feature selection (Wei et al., 2015b) and clustering (Wei et al., 2015a), demonstrating that respecting statistical dependencies between features improves downstream performance.

Our contributions are as follows. We introduce a novel self-supervised learning strategy that incorporates feature grouping using a conditional generative approach that uses variational autoencoders (Sohn et al., 2015) to synthesise realistic corrupted values conditioned on feature group membership. Next, we use three distinct evaluation settings: (1) cardiovascular health prediction where we pretrain on UK Biobank data and evaluate on community-gathered plaque assessment data, demonstrating real-world transfer learning capabilities and (2) 66 real-world tabular classification datasets from the OpenML-CC18 benchmark, and (3) 6 open source medical datasets, comparing against SCARF, multilayer perceptron (MLP), and random forest baselines. Our results demonstrate that explicitly modelling feature dependencies during pretraining yields representations that are more useful for downstream prediction tasks, particularly in low-label regimes where self-supervised pretraining provides the greatest benefit.

## 2. Methodology

### 2.1. Overview

Our framework extends contrastive self-supervised learning for tabular data by explicitly modeling feature dependencies through mutual information and generating realistic corruptions via conditional variational autoencoders. The key insight is that corrupting features whilst ignoring dependencies can create unrealistic samples that violate domain constraints, potentially degrading the quality of learned representations. Our approach identifies groups of statistically dependent features and learns to generate corruptions that preserve their realistic joint distributions whilst introducing sufficient variation for effective contrastive learning. Whilst

permutation-based corruption is limited to recombining existing feature values, a conditional variational auto-encoder (CVAE) (Sohn et al., 2015) learns the conditional distribution of feature groups and can generate novel realistic values beyond simple data resampling.

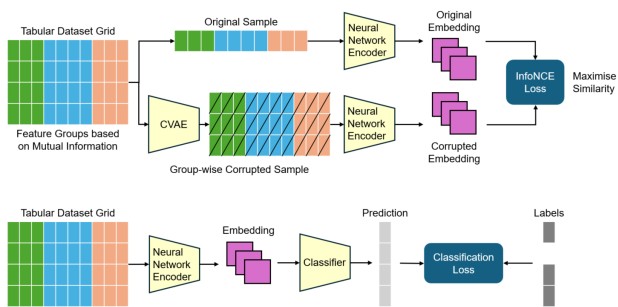

*Figure 1.* Overview of MI-guided grouped corruption for tabular self-supervised learning. Features are clustered based on mutual information (indicated by colour), and entire feature groups are corrupted coherently.

### 2.2. Mutual Information-Based Feature Clustering

For a dataset with $d$ features, we construct a pairwise Mutual Information (MI) matrix $\mathbf{M} \in \mathbb{R}^{d \times d}$ where $M_{ij} = I(X_i; X_j)$ quantifies the statistical dependence between features $i$ and $j$. We estimate mutual information using the k-nearest neighbours method (see Appendix A.2 for details on handling discrete, continuous, and mixed feature types) and normalise the MI matrix to $[0, 1]$ by dividing by the maximum observed MI value, yielding $\tilde{\mathbf{M}}$.

We convert $\tilde{\mathbf{M}}$ into a dissimilarity matrix $D_{ij} = 1 - \tilde{M}_{ij}$ and perform hierarchical agglomerative clustering with average linkage. Features are grouped based on a distance threshold $\sigma = 1 - \theta$, where $\theta$ is the MI similarity threshold hyperparameter. Features within the same cluster exhibit MI above $\theta$. We impose minimum and maximum cluster size constraints to ensure meaningful groups whilst preventing excessively large clusters.

### 2.3. Conditional Variational Auto-encoder for Structured Corruption

After feature grouping, we denote the set of feature clusters by $\mathcal{C} = \{C_1, C_2, \ldots, C_k\}$, where each $C_j \subseteq \{1, 2, \cdots, d\}$ represents feature indices in cluster $j$. For any sample $\mathbf{x}$, we denote its subvector on cluster $C_j$ as $\mathbf{x}_{C_j}$ and the complementary features as $\mathbf{x}_{-C_j}$.

For each cluster $C_j$, we train a CVAE with latent variable $\mathbf{z}$ to model the conditional distribution $p(\mathbf{x}_{C_j}|\mathbf{x}_{-C_j})$:

$$p_\phi(\mathbf{x}_{C_j}|\mathbf{x}_{-C_j}) = \int p_\phi(\mathbf{x}_{C_j}|\mathbf{z}, \mathbf{x}_{-C_j})p(\mathbf{z})\,d\mathbf{z} \quad (1)$$

where $\mathbf{z} \in \mathbb{R}^{16}$ and $p(\mathbf{z}) = \mathcal{N}(\mathbf{0}, \mathbf{I})$. The CVAE consists of an encoder $q_\psi(\mathbf{z}|\mathbf{x}_{C_j}, \mathbf{x}_{-C_j})$ and decoder $p_\phi(\mathbf{x}_{C_j}|\mathbf{z}, \mathbf{x}_{-C_j})$, both implemented as multilayer perceptrons with batch normalisation and ReLU activations (see Appendix A.3 for full architecture details).

We train each CVAE by minimising:

$$\begin{aligned}
\mathcal{L}^j_{\text{CVAE}} = &-\mathbb{E}_{q_\psi}\left[\log p_\phi(\mathbf{x}_{C_j} \mid \mathbf{z}, \mathbf{x}_{-C_j})\right] \\
&+ \beta \cdot D_{\text{KL}}\left(q_\psi(\mathbf{z} \mid \mathbf{x}_{C_j}, \mathbf{x}_{-C_j}) \,\|\, p(\mathbf{z})\right)
\end{aligned} \quad (2)$$

where $\beta = 0.01$ weights the KL term.

### 2.4. Marginal Sampling and Corruption Generation

To avoid trivial reconstructions, we condition the CVAE on a perturbed version of $\mathbf{x}_{-C_j}$ rather than the exact conditioning features from the same sample. We maintain empirical marginal pools $\{\mathcal{P}_i\}_{i=1}^d$ by pre-sampling maximum 10,000 values per feature from the training data. During corruption, we select one conditioning feature uniformly at random and replace its value with a sample from the corresponding marginal pool, creating a counterfactual conditioning vector $\tilde{\mathbf{x}}_{-C_j}$ that preserves most dependencies whilst breaking the joint distribution.

During corruption, for each sample in a mini-batch, we independently corrupt each feature group $C_j$ with probability $p_{\text{corrupt}}$. For selected groups, we construct $\tilde{\mathbf{x}}_{-C_j}$ by randomly perturbing one feature as described above, sample $\mathbf{z} \sim \mathcal{N}(\mathbf{0}, \mathbf{I})$, then generate $\tilde{\mathbf{x}}_{C_j} \sim p_\phi(\mathbf{x}_{C_j} \mid \mathbf{z}, \tilde{\mathbf{x}}_{-C_j})$ to replace the original subvector. This single-feature perturbation ensures stable corruption patterns independent of batch composition. Figure 1 illustrates this pipeline.

### 2.5. Contrastive Learning Objective

Following SCARF (Bahri et al., 2022), we train the encoder using InfoNCE loss (Oord et al., 2018). For each sample $\mathbf{x}_i$ and its corrupted view $\tilde{\mathbf{x}}_i$, we compute encoder representations $\mathbf{h}_i = f_\theta(\mathbf{x}_i)$ and $\tilde{\mathbf{h}}_i = f_\theta(\tilde{\mathbf{x}}_i)$, followed by $\ell_2$-normalised projected embeddings:

$$\mathbf{q}_i = \frac{g_\omega(\mathbf{h}_i)}{\|g_\omega(\mathbf{h}_i)\|_2}, \qquad \tilde{\mathbf{q}}_i = \frac{g_\omega(\tilde{\mathbf{h}}_i)}{\|g_\omega(\tilde{\mathbf{h}}_i)\|_2} \quad (3)$$

The contrastive loss for positive pair $(i, \text{pos}(i))$ is:

$$\ell_{i,\text{pos}(i)} = -\log\frac{\exp(\text{sim}(\mathbf{q}_i, \mathbf{q}_{\text{pos}(i)})/\tau)}{\sum_{k=1}^{2N}\mathbf{1}_{[k\neq i]}\exp(\text{sim}(\mathbf{q}_i, \mathbf{q}_k)/\tau)} \quad (4)$$

where $\text{sim}(\mathbf{u}, \mathbf{v}) = \mathbf{u}^\top\mathbf{v}$ and $\tau = 0.07$. The final loss averages over all positive pairs: $\mathcal{L}_{\text{InfoNCE}} = \frac{1}{2N}\sum_{i=1}^{2N}\ell_{i,\text{pos}(i)}$.

### 2.6. Implementation Details

The encoder $f_\theta$ is a four-layer MLP with hidden dimension 256, batch normalisation, and ReLU activations. The projection head $g_\phi$ is a two-layer MLP discarded after pretraining. We set MI threshold $\theta = 0.3$ with minimum cluster size 2 and maximum 50% of features. Each CVAE uses hidden dimension 128, trained for 100 epochs with Adam (learning rate $10^{-3}$). Contrastive pretraining uses Adam with learning rate $10^{-3}$, weight decay $10^{-5}$, and $p_{\text{corrupt}} = 0.6$.

## 3. Results

We evaluate our mutual information-guided grouped corruption (MI-CVAE) approach across two experimental settings: cardiovascular health prediction using large-scale biobank pretraining with small community-gathered evaluation data, and a diverse benchmark of real-world classification tasks. As an ablation, we also compare against the simpler method of using the clusters found in Section 2.2 and randomly swapping these in blocks (MI-Grouped). Our results demonstrate that MI-guided feature grouping consistently improves label efficiency compared to existing self-supervised and supervised baselines, requiring substantially fewer labelled examples to achieve comparable performance.

### 3.1. Cardiovascular Health Prediction: Transfer Learning from UK Biobank to Community Plaque Data

We evaluate the effectiveness of our approach in a realistic medical scenario where large-scale unlabelled data is available for pretraining but downstream evaluation must be performed on small, community-gathered datasets. We pretrain all self-supervised methods on 70,000 participants from the UK Biobank with no labels, then evaluate on plaque presence prediction using 300 participants from community cardiovascular screening events in Hong Kong. This experimental setup reflects the common real-world situation where large hospital or research datasets lack labels for specific conditions that are only available in smaller specialised cohorts.

Table 1 presents the results of 5-fold cross-validation on the community plaque dataset after pretraining on UK Biobank data. Our MI-CVAE grouped corruption approach achieves substantial improvements over all baselines across multiple metrics. Compared to the standard MLP baseline trained only on the labelled community data, MI-CVAE improves accuracy by 16.8% (0.936 vs 0.802), F1 score by 71.8% (0.768 vs 0.447), and AUC by 22.3% (0.938 vs 0.767). Comparing against SCARF, which also benefits from UK

*Table 1.* Performance on community plaque detection after pretraining on UK Biobank data.

| METHOD | ACCURACY | PRECISION | RECALL | F1 | AUC |
|---|---|---|---|---|---|
| MLP | 0.802 ± 0.036 | 0.457 ± 0.073 | 0.446 ± 0.073 | 0.447 ± 0.061 | 0.767 ± 0.039 |
| RANDOM FOREST | 0.821 ± 0.039 | 0.460 ± 0.408 | 0.159 ± 0.142 | 0.228 ± 0.198 | 0.808 ± 0.040 |
| TABPFN | 0.828 ± 0.033 | 0.603 ± 0.185 | 0.232 ± 0.042 | 0.329 ± 0.071 | 0.841 ± 0.061 |
| SCARF | 0.908 ± 0.046 | 0.610 ± 0.310 | 0.692 ± 0.358 | 0.648 ± 0.330 | 0.894 ± 0.123 |
| **MI-CVAE** | **0.936 ± 0.050** | **0.833 ± 0.087** | **0.765 ± 0.308** | **0.768 ± 0.247** | **0.938 ± 0.079** |

Biobank pretraining but uses independent feature corruption, MI-CVAE achieves improvements: 3.2% higher accuracy (0.936 vs 0.908), 18.6% higher F1 score (0.768 vs 0.648), and 5.0% higher AUC (0.938 vs 0.894).

### 3.2. Open Source Dataset Performance

To assess the generalisability of our approach, we evaluate on an extended version of the OpenML-CC18 benchmark comprising 66 diverse real-world classification tasks (see Appendix A for complete list). The original OpenML-CC18 curated collection spans domains including finance, science, marketing, and engineering. We augment this benchmark with six additional medical datasets covering cardiovascular health, diabetes prediction, cancer diagnosis, and other clinical applications, due to the high mutual information between features in typical medical datasets. We exclude three image datasets (MNIST, Fashion-MNIST, and CIFAR-10) from the original collection as they are not tabular data. We assess using 5-fold cross-validation for each dataset, with 80% training data and 20% test data for each fold.

Figure 2 presents comprehensive performance across all 72 datasets and label fractions, revealing consistent patterns in method behaviour. MI-guided approaches consistently outperform random corruption and supervised baselines across low label fractions. At 1% and 5% label fractions MI-CVAE has a higher F1 score across 77.7% and 75% of datasets respectively. The advantage is most pronounced at low label fractions: at 1% labels, MI-CVAE achieves 0.630 accuracy compared to 0.549 for SCARF and 0.587 for supervised MLP, whilst at 5% labels the gap widens with MI-CVAE reaching 0.780 accuracy and 0.914 AUC versus 0.753 and 0.878 for SCARF. The performance gains persist through intermediate label fractions, with MI-CVAE maintaining leads of 1-3 percentage points in accuracy over SCARF at 10% and 25% labels. MI-Grouped corruption follows closely behind MI-CVAE, consistently outperforming random corruption. At higher label fractions (50% and above), performance differences narrow as all methods converge, confirming that sufficient labeled data allows all approaches to learn effective representations. However, the structured corruption methods achieve comparable performance with substantially fewer labels, demonstrating superior label efficiency (see Appendix A.6 for complete results).

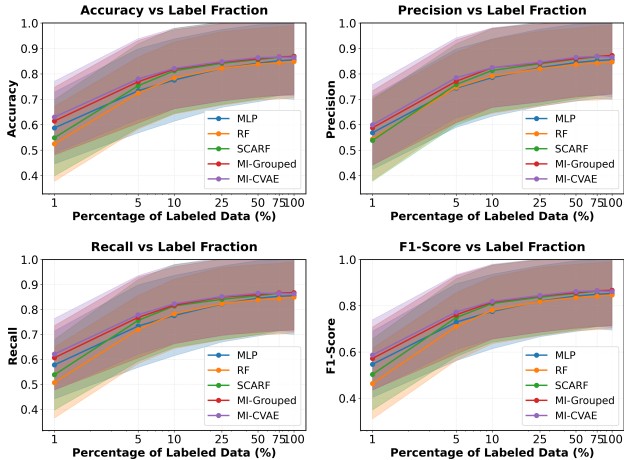

*Figure 2.* Label efficiency curves showing performance as a function of labelled training data fraction. MI-CVAE (purple) and MI-Grouped (orange) demonstrate superior label efficiency compared to random corruption (green) and supervised baselines, particularly at low label fractions (1-10%).

## 4. Conclusion

We introduced a MI guided approach to self-supervised learning for tabular data that respects the semantic structure encoded in feature dependencies. By identifying groups of related features through MI estimation and applying coherent corruptions within these groups, our method produces representations that preserve meaningful relationships whilst providing effective contrastive signals for pretraining. Evaluation on plaque prediction and 66 diverse classification tasks demonstrates consistent improvements in label efficiency. These results validate our central hypothesis that self-supervised learning for tabular data should account for feature dependencies rather than treating all features as independent. For domains like healthcare where large unlabelled datasets are abundant but labelled data requires expert annotation, this improved label efficiency translates directly to reduced annotation costs and faster model development. Future work could explore learned corruption strategies that adapt to local feature relationships, and investigate how structured corruption interacts with other self-supervised objectives beyond contrastive learning.

## Acknowledgements

This study was supported by the InnoHK initiative of the Innovation and Technology Commission of the Hong Kong Special Administrative Region Government.

## Impact Statement

This paper presents work whose goal is to advance the field of Machine Learning, with a focus on automated techniques for screening and diagnosis. By enabling faster, scalable, and cost-effective assessments, such methods have the potential to make healthcare more accessible, especially in underserved or resource-limited settings.

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

# A. Dataset Details

## A.1. OpenML-CC18 Datasets

We use 75 classification tasks from OpenML with the following dataset IDs: 3, 6, 11, 12, 14, 15, 16, 18, 22, 23, 28, 29, 31, 32, 37, 44, 46, 50, 54, 151, 182, 188, 38, 307, 300, 458, 469, 1049, 1050, 1053, 1063, 1067, 1068, 1590, 4134, 1510, 1489, 1494, 1497, 1501, 1480, 1485, 1486, 1487, 1468, 1475, 1462, 1464, 4534, 6332, 1461, 4538, 1478, 23381, 40499, 40668, 40966, 40982, 40994, 40983, 40975, 40984, 40979, 41027, 23517, 40923, 40978, 40670, 40701, 57, 329, 1466, 43227, 43469, 43672. These exclude the three image datasets (MNIST, Fashion-MNIST, CIFAR-10) from the original OpenML-CC18 suite.

## A.2. Mutual Information Estimation Details

Mutual information (MI) quantifies the statistical dependence between two random variables. For discrete feature pairs, MI is defined in terms of the probability mass function (PMF) as:

$$I(X_i; X_j) = \sum_{x_i, x_j} p(x_i, x_j) \log \frac{p(x_i, x_j)}{p(x_i)p(x_j)} \tag{5}$$

For continuous feature pairs, MI is defined in terms of the probability density function (PDF) as:

$$I(X_i; X_j) = \int \int p(x_i, x_j) \log \frac{p(x_i, x_j)}{p(x_i)p(x_j)} \, dx_i \, dx_j \tag{6}$$

Since tabular datasets may contain both discrete and continuous features, MI estimation depends on feature type. We estimate mutual information using the k-nearest neighbours method, which provides consistent estimates for both discrete and continuous variables. The resulting MI matrix is normalised to the range $[0, 1]$ by dividing each entry by the maximum observed MI value, yielding a normalized dependency matrix $\tilde{\mathbf{M}}$. This normalization facilitates interpretable threshold selection.

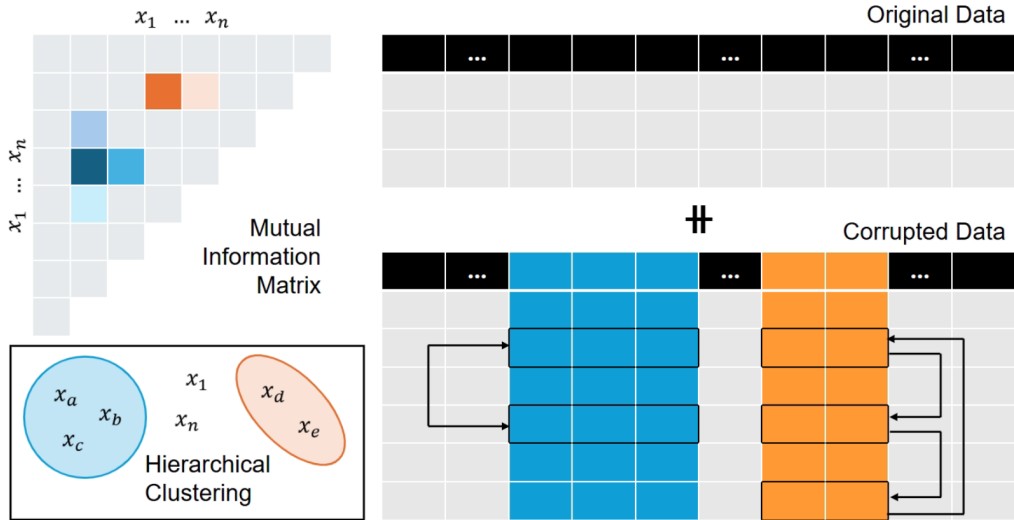

*Figure 3.* Example mutual information matrix showing feature dependencies. Features with high mutual information (indicated by brighter colours) are grouped together using hierarchical clustering.

Figure 3 shows an example MI matrix computed from real tabular data, illustrating how features naturally group into clusters based on their statistical dependencies.

## A.3. Network Architecture Details

### A.3.1. CVAE ARCHITECTURE

Each CVAE consists of an encoder network and a decoder network. The encoder network $q_\psi(\mathbf{z}|\mathbf{x}_{C_j}, \mathbf{x}_{-C_j})$ takes as input the concatenation of cluster features and complementary features, and outputs parameters of a Gaussian distribution over the latent space:

- Input layer: dimension $|C_j| + |-C_j|$

- Hidden layer 1: 128 units, batch normalisation, ReLU activation

- Hidden layer 2: 128 units, batch normalisation, ReLU activation

- Output layer: 32 units (16 for mean, 16 for log-variance)

The decoder network $p_\phi(\mathbf{x}_{C_j}|\mathbf{z}, \mathbf{x}_{-C_j})$ takes as input the concatenation of the latent code and complementary features, and reconstructs the cluster features:

- Input layer: dimension $16 + |-C_j|$

- Hidden layer 1: 128 units, batch normalisation, ReLU activation

- Hidden layer 2: 128 units, batch normalisation, ReLU activation

- Output layer: $|C_j|$ units (reconstructed cluster features)

We use the reparameterisation trick for backpropagation through the stochastic latent variable: $\mathbf{z} = \boldsymbol{\mu} + \boldsymbol{\sigma} \odot \boldsymbol{\epsilon}$ where $\boldsymbol{\epsilon} \sim \mathcal{N}(\mathbf{0}, \mathbf{I})$.

### A.3.2. ENCODER AND PROJECTION HEAD ARCHITECTURE

The encoder $f_\theta$ processes the full feature vector:

- Input layer: dimension $d$ (number of features)

- Hidden layer 1: 256 units, batch normalisation, ReLU activation

- Hidden layer 2: 256 units, batch normalisation, ReLU activation

- Hidden layer 3: 256 units, batch normalisation, ReLU activation

- Hidden layer 4: 256 units, batch normalisation, ReLU activation

- Output: 256-dimensional representation

The projection head $g_\omega$ maps encoder outputs to the embedding space for contrastive learning:

- Hidden layer: 256 units, batch normalisation, ReLU activation

- Output layer: 128 units (embedding dimension)

Following standard practice in contrastive learning (Chen et al., 2020), the projection head is discarded after pretraining, and only the encoder representations are used for downstream tasks.

---

**Algorithm 1** CVAE-Based Feature Corruption

---

**Require:** Mini-batch $\mathbf{X} \in \mathbb{R}^{B \times d}$, feature clusters $\mathcal{C} = \{C_1, \ldots, C_k\}$, trained CVAEs $\{\text{CVAE}_1, \ldots, \text{CVAE}_k\}$, marginal pools $\{\mathcal{P}_i\}_{i=1}^d$ of size $N_{\text{pool}}$ per feature, corruption probability $p_{\text{corrupt}}$
**Ensure:** Corrupted batch $\tilde{\mathbf{X}} \in \mathbb{R}^{B \times d}$
1: Initialize $\tilde{\mathbf{X}} \leftarrow \mathbf{X}$
2: **for** each cluster $C_j \in \mathcal{C}$ **do**
3:     Sample corruption mask mask $\sim \text{Bernoulli}(p_{\text{corrupt}}, B)$ with $n_{\text{corrupt}} = \sum \text{mask}$
4:     **if** $n_{\text{corrupt}} > 0$ **then**
5:         Define other $\leftarrow \{1, \ldots, d\} \setminus C_j$ and extract $\mathbf{X}_{\text{cond}} \leftarrow \mathbf{X}[:, \text{other}]$
6:         Select feature to perturb: $k \sim \text{Uniform}(0, |\text{other}|)$
7:         Sample pool indices: idx $\sim \text{Uniform}(0, N_{\text{pool}}, n_{\text{corrupt}})$
8:         Perturb: $\mathbf{X}_{\text{cond}}[\text{mask}, k] \leftarrow \mathcal{P}_{\text{other}[k]}[\text{idx}]$
9:         Sample latent codes: $\mathbf{z} \sim \mathcal{N}(\mathbf{0}, \mathbf{I}_\ell)$ where $\ell$ is latent dimension
10:         Generate: $\tilde{\mathbf{X}}_{\text{cluster}} \leftarrow \text{CVAE}_j.\text{decode}(\mathbf{z}, \mathbf{X}_{\text{cond}})$
11:         Apply corruption: $\tilde{\mathbf{X}}[\text{mask}, C_j] \leftarrow \tilde{\mathbf{X}}_{\text{cluster}}[\text{mask}, :]$
12:     **end if**
13: **end for**
14: **return** $\tilde{\mathbf{X}}$

---

### A.4. Marginal Distribution Estimation

### A.5. CVAE Corruption Algorithm

The algorithm operates in three phases per cluster: first, conditioning features are perturbed by replacing a single randomly selected feature with a value sampled from its marginal pool $\mathcal{P}_i$ (line 8), ensuring that the perturbation breaks the joint distribution whilst maintaining feature-wise realism. Second, the CVAE generates what the cluster features should be given these perturbed conditions by sampling from the prior $p(\mathbf{z})$ and decoding with the perturbed context (lines 9–10). Finally, the generated values replace the original cluster features only for samples selected by the corruption mask (line 11). The marginal pools $\{\mathcal{P}_i\}_{i=1}^d$ are constructed during training by sampling $N_{\text{pool}} = 10,000$ values per feature from the training data, providing $\mathcal{O}(1)$ lookup complexity and corruption patterns independent of batch composition. Computational complexity is $\mathcal{O}(B \cdot \sum_j |C_j|)$ per batch for CVAE forward passes, with space complexity $\mathcal{O}(d \cdot N_{\text{pool}})$ for storing marginal pools.

### A.6. Detailed Benchmark Results

Table 2 presents comprehensive performance statistics across all 66 datasets and label fractions, revealing consistent patterns in method behaviour. The results demonstrate that MI-guided grouped corruption provides systematic improvements in label efficiency, with advantages most pronounced at low supervision levels where pretraining quality has the greatest impact. This experiment took approximately 11 hours to run on a PC with an Intel I9-13900K 24 core processor, with 64GB RAM and a NVIDIA 4080 graphics card.

The standard deviations across datasets reveal substantial task heterogeneity, with accuracy standard deviations ranging from 0.141 to 0.167 depending on label fraction and method. This variation reflects the diversity of the benchmark, which includes datasets from multiple domains with different feature types, sample sizes, and class distributions. The consistency of MI-guided improvements across this heterogeneous collection suggests that structured corruption provides robust benefits regardless of specific task characteristics.

### A.7. Comparison against TabPFN

We evaluated our approach against TabPFN 2.5 (Grinsztajn et al., 2025), a state-of-the-art method for tabular classification that uses in-context learning with pretrained transformers. TabPFN has demonstrated strong performance on small to medium-sized tabular datasets by learning priors over entire dataset distributions rather than individual samples.

*Table 2.* Performance across 66 OpenML-CC18 benchmark datasets at varying label fractions. Results show mean ± standard deviation across all datasets with 5-fold cross-validation. MI-CVAE consistently outperforms baselines at low to moderate label fractions.

| LABEL FRACTION | METHOD | ACCURACY | PRECISION | RECALL | F1 | AUC |
|---|---|---|---|---|---|---|
| 1% | MLP | 0.587 ± 0.141 | 0.568 ± 0.144 | 0.578 ± 0.136 | 0.547 ± 0.143 | **0.805 ± 0.108** |
| | RANDOM FOREST | 0.525 ± 0.147 | 0.544 ± 0.162 | 0.507 ± 0.142 | 0.463 ± 0.153 | 0.207 ± 0.347 |
| | SCARF | 0.549 ± 0.150 | 0.538 ± 0.160 | 0.538 ± 0.141 | 0.503 ± 0.154 | 0.706 ± 0.194 |
| | MI-GROUPED | 0.615 ± 0.133 | 0.588 ± 0.146 | 0.606 ± 0.129 | 0.572 ± 0.136 | 0.703 ± 0.176 |
| | MI-CVAE | **0.630 ± 0.141** | **0.600 ± 0.158** | **0.621 ± 0.143** | **0.587 ± 0.153** | 0.668 ± 0.191 |
| 5% | MLP | 0.733 ± 0.165 | 0.744 ± 0.154 | 0.733 ± 0.166 | 0.729 ± 0.167 | 0.891 ± 0.114 |
| | RANDOM FOREST | 0.724 ± 0.142 | 0.747 ± 0.138 | 0.720 ± 0.136 | 0.710 ± 0.147 | 0.852 ± 0.146 |
| | SCARF | 0.753 ± 0.165 | 0.758 ± 0.160 | 0.756 ± 0.164 | 0.749 ± 0.168 | 0.878 ± 0.134 |
| | MI-GROUPED | 0.769 ± 0.163 | 0.771 ± 0.163 | 0.768 ± 0.162 | 0.761 ± 0.169 | 0.908 ± 0.116 |
| | MI-CVAE | **0.780 ± 0.159** | **0.785 ± 0.157** | **0.778 ± 0.158** | **0.773 ± 0.162** | **0.914 ± 0.111** |
| 10% | MLP | 0.776 ± 0.162 | 0.786 ± 0.152 | 0.776 ± 0.161 | 0.775 ± 0.161 | 0.911 ± 0.117 |
| | RANDOM FOREST | 0.784 ± 0.142 | 0.791 ± 0.146 | 0.782 ± 0.139 | 0.780 ± 0.144 | 0.910 ± 0.117 |
| | SCARF | 0.812 ± 0.167 | 0.813 ± 0.165 | 0.814 ± 0.166 | 0.810 ± 0.168 | 0.927 ± 0.120 |
| | MI-GROUPED | 0.818 ± 0.156 | **0.824 ± 0.155** | 0.817 ± 0.155 | 0.815 ± 0.159 | 0.932 ± 0.111 |
| | MI-CVAE | **0.821 ± 0.158** | **0.824 ± 0.157** | **0.822 ± 0.158** | **0.819 ± 0.161** | **0.935 ± 0.108** |
| 25% | MLP | 0.822 ± 0.153 | 0.824 ± 0.152 | 0.822 ± 0.152 | 0.820 ± 0.154 | 0.935 ± 0.102 |
| | RANDOM FOREST | 0.821 ± 0.145 | 0.820 ± 0.147 | 0.823 ± 0.145 | 0.819 ± 0.148 | 0.941 ± 0.087 |
| | SCARF | 0.841 ± 0.162 | 0.840 ± 0.162 | 0.841 ± 0.161 | 0.837 ± 0.164 | 0.937 ± 0.115 |
| | MI-GROUPED | 0.846 ± 0.152 | 0.843 ± 0.153 | 0.849 ± 0.152 | 0.843 ± 0.156 | 0.943 ± 0.103 |
| | MI-CVAE | **0.848 ± 0.156** | **0.846 ± 0.157** | **0.851 ± 0.157** | **0.844 ± 0.159** | **0.945 ± 0.102** |
| 50% | MLP | 0.842 ± 0.151 | 0.844 ± 0.151 | 0.845 ± 0.151 | 0.842 ± 0.153 | 0.944 ± 0.096 |
| | RANDOM FOREST | 0.838 ± 0.141 | 0.837 ± 0.143 | 0.839 ± 0.141 | 0.836 ± 0.144 | 0.949 ± 0.083 |
| | SCARF | 0.856 ± 0.156 | 0.858 ± 0.157 | 0.855 ± 0.156 | 0.854 ± 0.158 | 0.946 ± 0.101 |
| | MI-GROUPED | 0.859 ± 0.151 | 0.860 ± 0.151 | 0.859 ± 0.152 | 0.857 ± 0.154 | 0.948 ± 0.100 |
| | MI-CVAE | **0.864 ± 0.152** | **0.865 ± 0.152** | **0.865 ± 0.152** | **0.861 ± 0.155** | **0.950 ± 0.096** |
| 100% | MLP | 0.855 ± 0.156 | 0.856 ± 0.157 | 0.855 ± 0.157 | 0.853 ± 0.158 | 0.952 ± 0.094 |
| | RANDOM FOREST | 0.849 ± 0.142 | 0.847 ± 0.144 | 0.849 ± 0.142 | 0.847 ± 0.144 | **0.955 ± 0.079** |
| | SCARF | **0.870 ± 0.149** | **0.872 ± 0.151** | **0.867 ± 0.148** | **0.867 ± 0.152** | 0.953 ± 0.090 |
| | MI-GROUPED | 0.868 ± 0.150 | **0.872 ± 0.151** | **0.867 ± 0.150** | **0.867 ± 0.153** | 0.952 ± 0.094 |
| | MI-CVAE | 0.865 ± 0.151 | 0.866 ± 0.156 | 0.863 ± 0.151 | 0.862 ± 0.156 | 0.953 ± 0.094 |

### A.7.1. EXPERIMENTAL SETUP

For a fair comparison, we selected datasets from the OpenML-CC18 benchmark where TabPFN could produce valid outputs and had not been used during TabPFN's pretraining phase, so we could not use the entire OpenML-CC18 benchmark for this comparison. This resulted in 11 datasets: balance-scale (ID: 11), breast-w (ID: 15), cmc (ID: 23), credit-approval (ID: 29), credit-g (ID: 31), kr-vs-kp (ID: 3), mfeat-fourier (ID: 14), mfeat-karhunen (ID: 16), mfeat-morphological (ID: 18), mfeat-zernike (ID: 22), and optdigits (ID: 28). All methods were evaluated using identical train-test splits and 5-fold cross-validation at seven label fractions (1%, 5%, 10%, 25%, 50%, 75%, and 100%).

### A.7.2. TABPFN COMPARISON RESULTS

Table 3 presents aggregate performance across all 11 datasets at varying label fractions. Figure 4 visualizes the label efficiency curves for accuracy, precision, recall, and F1-score.

At very low label fractions (1%), MI-CVAE significantly outperforms all baselines, achieving an F1-score of 0.673 compared to TabPFN's 0.632 and SCARF's 0.598. This advantage persists through moderate label fractions, with MI-CVAE maintaining superior performance at 5% and 10% labeled data. The self-supervised pretraining enabled by our mutual information-aware corruption strategy proves particularly valuable when labeled data is scarce.

However, as label fraction increases beyond 25%, TabPFN's performance improves more rapidly, ultimately surpassing our approach at higher label fractions (50%-100%). This is expected, as TabPFN's in-context learning mechanism is

*Table 3.* Performance across 11 OpenML-CC18 benchmark datasets at varying label fractions. Results show mean $\pm$ standard deviation across all datasets with 5-fold cross-validation. MI-CVAE consistently outperforms baselines at low to moderate label fractions.

| LABEL FRACTION | METHOD | ACCURACY | PRECISION | RECALL | F1 | AUC |
|---|---|---|---|---|---|---|
| 1% | TABPFN | $0.649 \pm 0.140$ | $0.655 \pm 0.167$ | $0.633 \pm 0.129$ | $0.632 \pm 0.150$ | $0.825 \pm 0.158$ |
| | SCARF | $0.623 \pm 0.126$ | $0.627 \pm 0.146$ | $0.595 \pm 0.145$ | $0.598 \pm 0.143$ | $0.805 \pm 0.143$ |
| | MI-GROUPED | $0.671 \pm 0.164$ | $0.674 \pm 0.177$ | $0.674 \pm 0.158$ | $0.661 \pm 0.169$ | $0.825 \pm 0.154$ |
| | MI-CVAE | $\mathbf{0.688 \pm 0.170}$ | $\mathbf{0.697 \pm 0.183}$ | $\mathbf{0.676 \pm 0.165}$ | $\mathbf{0.673 \pm 0.176}$ | $\mathbf{0.834 \pm 0.147}$ |
| 5% | TABPFN | $0.758 \pm 0.148$ | $0.742 \pm 0.177$ | $0.739 \pm 0.185$ | $0.734 \pm 0.184$ | $\mathbf{0.896 \pm 0.131}$ |
| | SCARF | $0.755 \pm 0.149$ | $0.746 \pm 0.181$ | $0.731 \pm 0.180$ | $0.734 \pm 0.180$ | $0.876 \pm 0.141$ |
| | MI-GROUPED | $0.770 \pm 0.149$ | $0.765 \pm 0.167$ | $0.759 \pm 0.165$ | $0.755 \pm 0.168$ | $0.893 \pm 0.121$ |
| | MI-CVAE | $\mathbf{0.771 \pm 0.152}$ | $\mathbf{0.766 \pm 0.171}$ | $\mathbf{0.765 \pm 0.162}$ | $\mathbf{0.758 \pm 0.168}$ | $0.892 \pm 0.125$ |
| 10% | TABPFN | $0.799 \pm 0.138$ | $0.789 \pm 0.170$ | $0.788 \pm 0.161$ | $0.782 \pm 0.167$ | $\mathbf{0.918 \pm 0.112}$ |
| | SCARF | $0.796 \pm 0.152$ | $0.785 \pm 0.180$ | $0.785 \pm 0.164$ | $0.781 \pm 0.173$ | $0.901 \pm 0.129$ |
| | MI-GROUPED | $0.799 \pm 0.151$ | $0.789 \pm 0.179$ | $0.794 \pm 0.164$ | $0.787 \pm 0.172$ | $0.911 \pm 0.119$ |
| | MI-CVAE | $\mathbf{0.806 \pm 0.148}$ | $\mathbf{0.797 \pm 0.175}$ | $\mathbf{0.802 \pm 0.155}$ | $\mathbf{0.796 \pm 0.165}$ | $0.910 \pm 0.119$ |
| 25% | TABPFN | $\mathbf{0.839 \pm 0.134}$ | $\mathbf{0.827 \pm 0.164}$ | $\mathbf{0.831 \pm 0.148}$ | $\mathbf{0.825 \pm 0.156}$ | $\mathbf{0.937 \pm 0.098}$ |
| | SCARF | $0.829 \pm 0.148$ | $0.818 \pm 0.172$ | $0.816 \pm 0.162$ | $0.813 \pm 0.168$ | $0.917 \pm 0.119$ |
| | MI-GROUPED | $0.831 \pm 0.144$ | $0.818 \pm 0.168$ | $0.828 \pm 0.155$ | $0.819 \pm 0.162$ | $0.924 \pm 0.111$ |
| | MI-CVAE | $0.831 \pm 0.148$ | $0.818 \pm 0.168$ | $0.826 \pm 0.153$ | $0.818 \pm 0.162$ | $0.922 \pm 0.114$ |
| 50% | TABPFN | $\mathbf{0.857 \pm 0.134}$ | $\mathbf{0.844 \pm 0.154}$ | $\mathbf{0.848 \pm 0.146}$ | $\mathbf{0.844 \pm 0.150}$ | $\mathbf{0.943 \pm 0.090}$ |
| | SCARF | $0.841 \pm 0.145$ | $0.829 \pm 0.165$ | $0.830 \pm 0.153$ | $0.826 \pm 0.161$ | $0.922 \pm 0.115$ |
| | MI-GROUPED | $0.840 \pm 0.151$ | $0.824 \pm 0.169$ | $0.836 \pm 0.159$ | $0.828 \pm 0.165$ | $0.930 \pm 0.108$ |
| | MI-CVAE | $0.842 \pm 0.148$ | $0.831 \pm 0.169$ | $0.831 \pm 0.153$ | $0.828 \pm 0.162$ | $0.926 \pm 0.110$ |
| 100% | TABPFN | $\mathbf{0.871 \pm 0.130}$ | $\mathbf{0.858 \pm 0.142}$ | $\mathbf{0.850 \pm 0.168}$ | $\mathbf{0.851 \pm 0.156}$ | $\mathbf{0.947 \pm 0.085}$ |
| | SCARF | $0.844 \pm 0.144$ | $0.831 \pm 0.164$ | $0.828 \pm 0.158$ | $0.827 \pm 0.162$ | $0.924 \pm 0.109$ |
| | MI-GROUPED | $0.847 \pm 0.151$ | $0.833 \pm 0.164$ | $0.830 \pm 0.183$ | $0.828 \pm 0.176$ | $0.931 \pm 0.109$ |
| | MI-CVAE | $0.847 \pm 0.145$ | $0.833 \pm 0.156$ | $0.827 \pm 0.181$ | $0.826 \pm 0.170$ | $0.930 \pm 0.105$ |

specifically designed to leverage the prior knowledge encoded in its pretrained transformer when sufficient labeled examples are available for effective few-shot learning.

Table 4 summarizes the head-to-head comparison between MI-CVAE and SCARF based on F1-scores across individual datasets. MI-CVAE achieves particularly strong win rates at low label fractions (90.9% at 1% and 10% labels), demonstrating the effectiveness of our approach in the few-shot regime. As expected, the advantage diminishes at higher label fractions, where the benefits of self-supervised pretraining become less pronounced.

*Table 4.* Win rates for MI-CVAE vs TabPFN based on F1-score across 11 datasets. A "win" indicates MI-CVAE achieved higher F1-score on that dataset.

| LABEL FRACTION | WINS | LOSSES | WIN RATE (%) |
|---|---|---|---|
| 1% | 10 | 1 | 90.9 |
| 5% | 9 | 2 | 81.8 |
| 10% | 10 | 1 | 90.9 |
| 25% | 8 | 3 | 72.7 |
| 50% | 5 | 6 | 45.5 |
| 75% | 4 | 7 | 36.4 |
| 100% | 6 | 5 | 54.5 |

Notably, both MI-CVAE and TabPFN substantially outperform SCARF across most settings, validating the importance of either capturing feature dependencies (our approach) or learning strong priors through meta-learning (TabPFN's approach) for tabular data representation learning. The convergence of all methods at 100% labels suggests that with full supervision, the choice of pretraining or meta-learning strategy becomes less critical, though TabPFN maintains a slight edge due to its strong inductive biases.

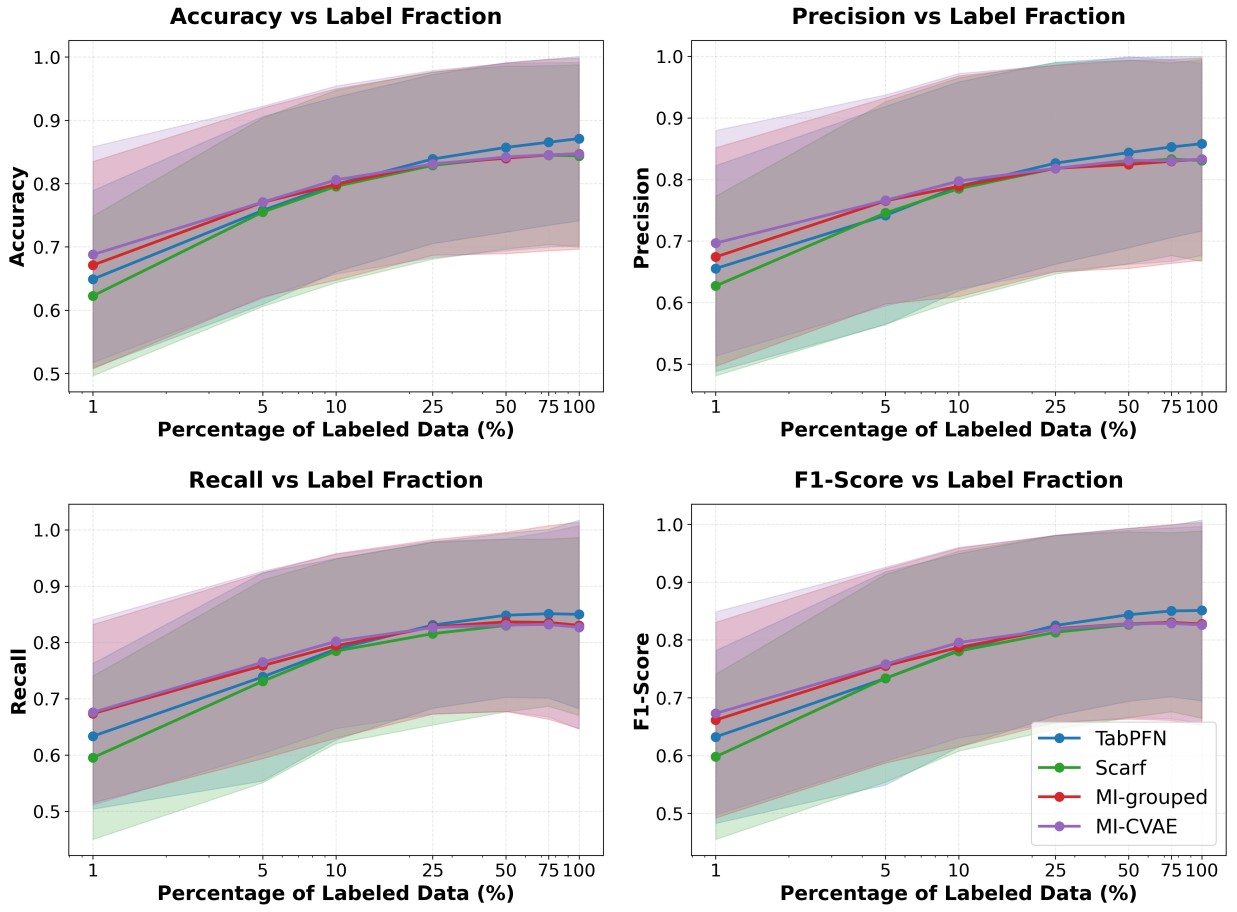

*Figure 4.* Label efficiency comparison between TabPFN, SCARF, MI-Grouped, and MI-CVAE across 11 OpenML-CC18 benchmark datasets. Shaded regions represent standard deviations. MI-CVAE demonstrates superior performance at low label fractions (1%-10%), with TabPFN gaining advantage at higher label fractions (50%-100%).

## A.8. Ablation Study Details

### A.8.1. MI THRESHOLD SENSITIVITY

Table 5 presents the complete results of our MI threshold sensitivity analysis conducted on a representative subset from the OpenML-CC18 benchmark (3, 6, 11, 12, 14, 15, 16, 18, 22, 23, 28, 29, 31, 32, 37). We evaluate thresholds from 0.15 to 0.5 to understand how the aggressiveness of feature grouping affects downstream performance.

The results demonstrate that the method exhibits robustness to threshold selection within a reasonable range. Thresholds between 0.2 and 0.4 produce consistently strong performance, varying by less than 2% in accuracy. This stability suggests the method is not overly sensitive to exact threshold tuning, which is desirable for practical deployment where optimal hyperparameters may vary across domains.

Very low thresholds (0.15) create excessive small clusters that fragment genuine dependencies. When the threshold is too permissive, even weakly dependent features are grouped together, resulting in clusters that do not capture meaningful semantic structure. Conversely, very high thresholds (0.5) produce overly large clusters that group weakly related features together, reducing the diversity of corruptions and limiting the contrastive signal.

Moderate thresholds around 0.3 provide the best trade-off between preserving dependencies and maintaining corruption diversity, consistently producing clusters that balance semantic coherence with sufficient independence for effective contrastive learning. The default threshold of 0.3 achieves near-optimal performance whilst remaining robust across diverse dataset characteristics.

*Table 5.* Effect of MI threshold on performance. Results show mean accuracy across 10 diverse datasets with 5-fold cross-validation. Performance remains stable across a wide range, with optimal values between 0.2 and 0.4.

| Threshold ($\theta$) | Mean Accuracy |
|---|---|
| 0.15 | 0.667 |
| 0.20 | 0.709 |
| 0.25 | 0.701 |
| 0.30 | 0.718 |
| 0.35 | 0.718 |
| 0.40 | 0.709 |
| 0.45 | 0.692 |
| 0.50 | 0.667 |

### A.8.2. DISTRIBUTION PRESERVATION ANALYSIS

To verify that performance improvements arise from exploiting feature dependencies rather than introducing distributional shifts, we measure how well MI-guided corruption preserves the original data distribution. We employ three complementary metrics:Kolmogorov-Smirnov (KS) Statistic, Jensen-Shannon (JS) Divergence, Wasserstein Distance.

*Table 6.* Mean distribution preservation metrics comparing corruption strategies across 10 representative datasets. Lower values indicate better preservation of the original distribution. All methods maintain nearly identical distributions.

| Method | KS Statistic | JS Divergence | Wasserstein |
|---|---|---|---|
| Random | 0.0423 | 0.1406 | 0.0576 |
| MI-Grouped | 0.0428 | 0.1410 | 0.0591 |
| MI-CVAE | 0.0427 | 0.1418 | 0.0581 |

Table 6 shows that all three corruption strategies maintain nearly identical distributions, with differences below 1% across all metrics. The MI-CVAE approach achieves Wasserstein distance comparable to random corruption (0.0581 vs 0.0576), indicating that structured corruption preserves the geometric properties of the data manifold whilst respecting feature dependencies.

The minimal divergence across methods demonstrates that label efficiency gains stem from coherent feature corruption rather than fortuitous distributional artifacts. This preservation is critical for semi-supervised learning, as it ensures the augmented samples remain representative of the true data distribution during self-supervised pretraining. The results confirm that our approach successfully maintains distributional fidelity whilst incorporating semantic structure through feature grouping.

