# OpenReview forum: "Mutual Information-Guided Corruption for Improved Self-Supervised Representation Learning in Tabular Data"
_ICML.cc/2026/Workshop/FMSD — FMSD @ ICML 2026 Poster_

### Official Review · Reviewer_VMZN · 2026-05-18
**Promising Direction but needs further investigation**

**Rating:** 5
**Confidence:** 3

**Review:**

Strengths:
-The idea of respecting feature dependencies during corruption is well motivated.
- The UK Biobank experiments presents a real world challenging dataset. Also, the evaluation across 72 datasets is exhaustive.
Improvement Areas:
-  The authors present a MI grouped ablation in table 2. This clearly shows that simple block swapping using the same MI cluster  has the same accuracy as the proposed MI-CVAE approach. The authors should
clarify the main contribution of the work.
- In table1 the standard deviations are quite large relative to improvements.
- State of the art baselines like VIME, SubTab have been discussed in the introduction but the proposed method has not been compared against them.
- The AUC results in table 2 are puzzling with MI-CVAE having the lowest AUC despite having the highest F1 score and accuracy. Is the dataset imbalanced?
Detailed Comments:
- Conduct additional ablation experiments to clearly identify the main contribution of the work.
- Present a breakdown of the increased computational overhead of the proposed MI-CVAE method compared to MI-Grouped.
- Compare with state of the art baselines.
The idea is promising but needs further improvement. I am currently placing it slightly below acceptance threshold because the main contribution is not clear.

---

### Official Review · Reviewer_zuqc · 2026-05-22
**Review for Paper ID 90**

**Rating:** 4
**Confidence:** 4

**Review:**

This paper proposes MI-CVAE, a framework for tabular data, which groups statistically related features and generates realistic corrupted views using per-group CVAEs. The core insight is that existing methods like SCARF corrupt features independently, ignoring inter-feature dependencies that are common in real-world tabular data.
Results: Evaluation on 66 datasets plus 6 medical tasks, MI-CVAE achieves high accuracy at 1% labels.

The advantages of this paper include: the motivation is principled, MI-based grouping is a non-parametric way to respect tabular structure, and the idea to generate realistic corruptions via CVAE rather than permutation is well-reasoned.
The evaluation scope is genuinely broad. The advantage in low-label settings is sometimes important for healthcare applications where labeled data is scarce.

I could still have several concerns as follows:

- Missing Ablation: A formal table is needed to separate gains from MI-based grouping vs. CVAE generation. Figure 2 suggests grouping drives most of the improvement.

- Performance Reversal: At 100% label fraction, SCARF outperforms MI-CVAE (F1 0.872 vs. 0.862). The paper must explain why the method underperforms when labels are abundant.

- MI Estimation Reliability: $k$-NN MI estimation is unreliable in high dimensions. The paper lacks analysis on dependency accuracy, sensitivity to $k$, and the risks of spurious groupings.

- Missing Efficiency Comparison: CVAE training adds significant overhead (11 hours on an NVIDIA 4080). A direct training cost comparison with SCARF is required.

- Hidden Hyperparameter Guidance: The critical clustering threshold ($\theta = 0.3$) lacks in-text sensitivity analysis and practical tuning advice.

While the method is well-motivated and empirically sound, the technical narrative is weakened by the missing ablation data, the high-label performance reversal, and the lack of MI reliability analysis. Addressing these points will significantly strengthen the submission.